# Consumer Preferences for Processed Meat Reformulation Strategies: A Prototype for Sensory Evaluation Combined with a Choice-Based Conjoint Experiment

Xinyi Hong [1,2], Chenguang Li [2], Liming Wang [3,4], Mansi Wang [1,*], Simona Grasso [2] and Frank J. Monahan [2]

1    School of Innovation and Entrepreneurship, Guangzhou University, Guangzhou 510006, China
2    School of Agriculture and Food Science, University College Dublin, D04V1W8 Dublin, Ireland
3    School of Economics and Management, Beijing University of Technology, Beijing 100124, China
4    Irish Institute for Chinese Studies, University College Dublin, D04V1W8 Dublin, Ireland
*    Correspondence: cywangmansi@gzhu.edu.cn

**Abstract:** Consumption trends demand healthier meat products and require research into reformulation strategies. Ambiguities in consumer preferences for two processed meat reformulation strategies (i.e., ingredient "reduction" and nutrient "addition") were investigated. Using physical prototypes of omega-3-enriched pork sausages and sensory evaluation to reduce hypothetical bias, followed by a choice-based conjoint experiment, results suggested that consumers valued both "addition" and "reduction" reformulation strategies, and consumers' willingness-to-pay (WTP) premiums were the highest for omega-3 addition, followed by fat reduction, and were lowest for salt reduction. Moreover, WTP was influenced by sensory preferences and was positively correlated with sensory liking levels. Providing health-related information improved consumers' sensory perceptions of omega-3-enriched sausages. Findings imply that reformulated healthier meat products are acceptable to consumers. Moreover, to enhance consumers' valuation on new launches of healthier processed meat products, meat manufacturers should inform consumers of health-related reformulation information, provide consumers with opportunities to taste newly developed healthier processed meat products, and continuously optimize consumers' sensory experience.

**Keywords:** reformulation strategy; healthier processed meat; sensory liking; consumer preferences; willingness-to-pay

## 1. Introduction

The worldwide market value for processed meat products is forecasted to increase from US$523.1 billion to US$737.2 billion over the period 2020–2026. In the U.S., processed meat products account for 22% of meat consumption [1], while in Europe, available data indicate that adults consume more processed meat than the recommended intake [2]. Nevertheless, processed meat products are often associated with low-quality dietary choices [3] as well as high salt and saturated fat intakes [4]. Reformulation and associated health-related claims provide potential solutions to overcome nutritional defects in processed meat products [5–7].

Generally, there are two strategies for reformulation, namely an "addition" strategy and a "reduction" strategy [1]. The "addition" strategy involves enriching processed meat products with substances with health benefits. Examples include fiber-enriched sausages [2] and omega-3 fatty acids enriched sausages [3]. The "reduction" strategy involves diminishing the content of unhealthy ingredients that are of concern. Examples include low-sodium dry fermented sausages [4], and nitrite reduced sausages and ham [5].

Nevertheless, consumer preferences towards "addition" and "reduction" strategies are ambiguous in the existing literature. On the one hand, some literature concludes that consumers place more value on the "reduction" strategy and little value on the "addition"

strategy. Schnettler, et al. [6] conducted a survey showing that consumers are willing to pay moderately more for reformulated frankfurter sausages with reduced sodium or saturated fat but not for those claiming to be fiber-enriched or cholesterol-reduced. A companion study indicated similar observations regarding perceived healthiness and purchasing intentions (i.e., much higher for reduced sodium/fat, slightly higher for fiber-enrichment and marginally higher for cholesterol reduction) [7]. Likewise, Shan, et al. [8] compared consumers' reaction to reduction in salt/fat with addition of nutrients (i.e., omega 3 and vitamin E) in several processed meat products (namely ham, sausages, and beef burgers) using rating-based conjoint analysis. They reported higher perceived healthiness and purchasing intentions for decreased salt and/or fat than for the added nutrients in processed meat products. Furthermore, Profeta, et al. [9] found that when consumers were asked to choose between regular meat burgers and hybrid meat burgers (hybrid meat products contain plant-based ingredients, not specific nutrients) via online questionnaires, the majority (59.4%) voted for the former option, whereas only 27.4% chose the latter. On the other hand, there are studies finding favorable opinions on the "addition" of nutrients into processed meat products. Zajac, Kulawik, Tkaczewska, Migdal and Pustkowiak [3] reported that consumers preferred sausages with 5% flaxseed addition over sausages with no addition regarding taste, appearance, tenderness, juiciness, smell and overall acceptability. Moreover, Neville, et al. [10] found no significant differences in sensory liking between plant protein-enriched beef burgers as well as pork sausages versus their commercial counterparts among consumers. Other consumer-accepted reformulated examples include fiber-added sausages [2], omega-3-enriched sausages [11], multi-nutrient enriched dry fermented sausages [12], plant-sterol enriched turkey [13], and plant-based hybrid beef burgers [14,15].

The inconsistency in findings may be due to different research methodologies employed. When using questionnaires only to describe hypothetically healthier processed meat products, consumers prefer a "reduction" strategy, but when a sensory evaluation of heathier meat prototypes is conducted, an "addition" strategy is likely to be also welcomed. In real situations, consumers may have purchased food products without having previously tasted them. Studying consumers' perceptions of novel food products is highly relevant to understanding their preferences and purchasing behaviors. Without real tasting, Konuk [16] finds that perceived taste has a significant impact on perceived quality, and perceived taste and perceived quality are both related to consumers' willingness-to-buy. Nevertheless, hypothetical bias is often a major cause for deviation in stated preference evaluation, causing participant overstatement or understatement [17]. The hypothetical bias problems tend to be intensified in the acceptance of novel meat products when there are few marketable products to refer to in consumers' minds.

In this context, a sensory evaluation helps to reduce hypothetical bias toward healthier meat products and a satisfactory tasting experience helps to overcome barriers for consumption [9]. A number of studies suggest that sensory evaluations are useful to moderate consumers' views of hypothetically healthier meat products and to reduce associated consumer skepticism [8,18,19]. These equivocal conclusions regarding consumers' preferences for "addition" or "reduction" reformulation strategies for healthier processed meat products may be resolved using a research design combining a sensory evaluation and a choice-based conjoint (CBC) experiment. Here, sensory experience can bridge the gap between a concept and a real product; the hypothetically constructed scenarios in a CBC experiment can measure consumers' willingness-to-pay (WTP) for both "addition" and "reduction" reformulation strategies.

For this purpose, a physical prototype of omega-3-enriched pork sausages, for sensory testing, was manufactured in a food laboratory. This study selected pork sausages as the base food carrier and omega-3 fatty acid as the enriched nutrient for a number of reasons. Irish pork sausages are common and widely consumed food products in Ireland [20]. Sausages are also globally popular, which ensures relevance for participants from various backgrounds and reduces bias due to food unfamiliarity. Sausages are suitable for the

addition of nutrients and/or the removal or replacement of ingredients during processing [21,22]. The formulation of direct addition of fish oil into processed meat products to alter the fatty acid composition was advocated by Decker and Park [23]. Furthermore, consumer preferences for omega-3-enriched processed meat (including pork sausages) were observed in some previous studies [8,24]. Therefore, omega-3-enriched sausages fulfil the perceived match between a food carrier and a healthier ingredient and avoid the negative influence of mismatched combinations [21]. Compared with omega-3-enriched sausages, low-fat and low-salt sausage products are more commonly seen at market. In this study, the salt content and fat content in sausages were not changed. Only omega-3-enriched sausages were manufactured as a prototype.

By incorporating a real sensory evaluation experience to reduce hypothetical bias, this study aimed to clarify existing ambiguities in consumer preferences and WTP towards two opposing reformulation strategies (i.e., nutrient "addition" and ingredient "reduction") in processed meat. Under this context, the sensory evaluation served two purposes: firstly, first-hand observing and tasting of the omega-3-enriched sausages decreased potential skepticism towards novel food products and allowed consumers to adjust to quality expectations and consumption intentions after experiencing the product [18]; secondly, sensory factors were taken into consideration to explain consumers' WTP. The objectives of this study were three-fold: (1) to measure consumers' sensory liking for a physical prototype of omega-3-enriched sausages; (2) to estimate consumers' WTP for omega-3-enriched sausages, reduced-fat sausages, and reduced-salt sausages; (3) to investigate how sensory liking influences the WTP for reformulated sausages. While this study addressed consumers' perceptions of two opposing processed meat reformulation strategies, a companion study by the authors investigated how different nutrition and health claim information influenced consumers' perceptions [11].

## 2. Methods

### 2.1. Omega-3 Enriched Sausages Preparation

The pork sausage meat batter was purchased from an established local butcher in Dublin (Fenelons, Stillorgan, Co. Dublin, Ireland). Cod liver oil (557 mg EPA and 472 mg DHA per 5 mL) was purchased from Holland & Barrett (Nutgrove Shopping Centre, Dublin 14, Ireland). Cod liver oil liquid (6 mL) was incorporated into the pork sausage meat (1 kg per batch) in a Stefan mixer (UMSK 5E–60E, Stephan Machinery GmbH, Stephanplatz 2, 31789 Hameln, Germany) at medium speed for 4 min, to give a projected level of 46.8 mg of EPA and DHA per 100 kcal of sausage meat (meeting the minimum requirement of 40 mg of the sum of EPA and DHA per 100 kcal of sausage meat to carry a claim regarding omega-3 fatty acids). Four batches of sausages (approximately 25 kg meat batter for each batch) were manufactured and were stuffed into collagen casings (Edicas NB Edible Casings, S.L., Spain) using a hydraulic sausage filler (Mainca EM12, Equipamientos Cárnicos, S.L., Spain) and then hand-linked into cocktail sausage size (70 mm length 22 mm diameter). The final weight of each cocktail sausage was approximately 10 g. Approximately 250 sausages were made in each batch.

### 2.2. Data Collection Procedure

This study was approved by the Human Research Ethics Committee for Sciences (reference number LS-17-91-Hong-Li) of University College Dublin (UCD). The data collection involved sensory evaluation data and CBC experimental data, collected in a sensory laboratory in compliance with ISO 8589 [25] at the UCD Institute of Food and Health. The process took on average 30 min per participant and was completed in one session. The questionnaire was administered using computers and on paper.

This study involved a total of 330 participants who were voluntarily recruited by snowball sampling on the UCD Belfield campus. They were all consumers of processed meat (e.g., sausages, nuggets, burgers, ham, bacon, salami, smoked meats) and of over 18 years old. Participants provided informed consent when participating in this study and

were not remunerated for their time. Out of 330, 326 participants were valid (see Section 3.1. for more details).

### 2.2.1. Sensory Evaluation

Sensory evaluations were completed in individual tasting booths in a sensory laboratory under artificial daylight type illumination and in a controlled temperature (22–24 degrees Celsius). The same participant evaluated three samples under two different conditions: participants tasted two sausages in the blind condition and one sausage in the informed condition. The sausage samples were presented in a sequential monadic order.

First, under a blind condition (with tasting and no information), two sensory evaluations of conventional pork sausages and omega-3-enriched pork sausages were conducted. Each time, participants evaluated a raw sausage sealed in a transparent bag for external appearance observation to imitate the situation on a shop shelf, and a cooked sausage, presented on an odorless white plastic plate with odorless wooden tooth picks for tasting. After observing the appearance of the raw sausage and tasting the cooked sausages, participants rated the appearance, taste, texture and overall liking in a 9-point structured scale, where 1 = "dislike extremely", 2 = "dislike very much", 3 = "dislike moderately", 4 = "dislike slightly", 5 = "neither like nor dislike", 6 = "like slightly", 7 = "like moderately", 8 = "like very much", and 9 = "like extremely". Conventional sausages and omega-3-enriched sausages were evaluated in randomized order and with three-digit random codes.

Second, participants self-assessed their familiarity with omega-3 on a structured 9-point scale where 1 = "extremely unfamiliar", 5 = "neither unfamiliar nor familiar", and 9 = "extremely familiar". After answering the familiarity question and before tasting (in the informed condition, below), all participants were informed that the sausages contained sufficient omega-3 fatty acids to be labeled with a European Food Safety Authority (EFSA) nutrition claim "source of omega-3 fatty acids".

Third, a sensory evaluation of omega-3 enriched pork sausages was conducted under the informed condition and only an omega-3-enriched sausage was given to each participant for evaluation, following the same evaluation procedure of the earlier blind condition.

### 2.2.2. Choice-Based Conjoint (CBC) Experiment

Following sensory evaluation, a CBC experiment was conducted, which collected information on participants' favored alternatives for hypothetical sausages with five attributes in varying combinations (Table 1). These five attributes and their associated levels were carefully specified to reflect the characteristics of critical value to consumers according to the preceding relevant literature [8,20,26]. Price and meat content were relevant market offerings, thus ensuring relevance to consumers in making their choices. The four price levels were €2.60, €2.80, €3.00 and €3.20 per 454 g pack of sausages, while the four meat content levels were 60%, 70%, 80% and 90% of pork meat in the sausages. The three claims, including the nutrient addition claim (i.e., "source of omega-3") and the two ingredient reduction claims (i.e., "reduced fat" and "reduced salt"), corresponding to both "addition" and "reduction" reformulation strategies, were of interest in this research.

A full factorial design of attributes and levels (Table 1) generated a total number of 128 alternatives ($4 \times 4 \times 2 \times 2 \times 2$). Considering the cognitive demand, these 128 alternatives were reduced to 24, and grouped into 12 choice tasks in the questionnaire. This procedure was achieved by using a modified Fedorov algorithm to ensure that the design was balanced and efficient [27,28].

Participants were asked to imagine that they had made a purchase of sausages and had indicated their most preferred alternative across all twelve choice tasks. Each task consisted of three alternatives: two hypothetically constructed sausage products and a "neither" option. The figures of choice tasks used in the choice experiment questionnaire were the same as in Hong, Li, Wang, Gao, Wang, Zhang and Monahan [11]. A no-purchase

option (i.e., neither) was added in each choice set, which allowed a participant to choose not to "buy" sausage products just like in a real shopping experience.

**Table 1.** Attributes of sausages used in choice experiment questionnaire.

| Attributes [a] | Levels | Notes |
|---|---|---|
| Price | Four:<br>€ 2.60, € 2.80, € 3.00, € 3.20 | Per pack price, net weight 454 g |
| Meat content | Four:<br>60%, 70%, 80%, 90% | Pork meat percentage in sausages |
| Nutrient enrichment claim | Two:<br>Source of Omega-3,<br>No Omega-3 | An eligible nutrition claim (as listed in the Annex to Regulation (EC) No. 1924/2006) of "source of omega-3 fatty acids" |
| Ingredient reduction claim | Two:<br>Reduced Fat,<br>No Fat Reduction | An eligible reduced (name of nutrient) claim (as listed in the Annex to Regulation (EC) No. 1924/2006) meaning "reduced in saturated fatty acids" |
| | Two:<br>Reduced Salt,<br>No Salt Reduction | An eligible reduced (name of nutrient) claim (as listed in the Annex to Regulation (EC) No. 1924/2006) meaning "reduced in sodium/salt". |

[a] The attributes of sausages used in choice experiment questionnaire were the same as in Hong, Li, Wang, Gao, Wang, Zhang and Monahan [11].

Lastly, participants were asked to answer additional attitudinal, demographic, and behavioral questions. The questions covered food consumption habits (e.g., indication of consumption frequency of processed meat) and socio-demographic characteristics (e.g., gender, age, education, employment, monthly food budget and income).

*2.3. Econometric Models and Data Analysis*

Observation of the choices made by participants manifests the consumer utility derived from each alternative, based on the Lancaster consumer theory and random utility theory (RUT) [29,30]. Suppose a consumer, denoted $n$, gets some utility $U_{njt}$ from each alternative. Then, the latent utility is further divided into two parts: one part is observed and constructed as a function of explainable variables related to the alternatives and the other part is unobserved and randomly varies among alternatives and consumers [30]. A mathematical denotation of utility is modeled as follows:

$$U_{njt} = V_{njt} + \varepsilon_{njt} \tag{1}$$

where $U_{njt}$ denotes the total utility obtained by a consumer $n$, from the alternative $j$ ($j = 1, \dots, J$) in the choice set $t$. $V_{njt}$ measures utility by a vector of explainable variables constructed by researcher. $\varepsilon_{njt}$ represents the difference between the measured utility and the total utility. Under the utility-maximization assumption, a consumer $n$ chooses an alternative $i$ among all J alternatives within the same choice set $t$, if and only if $U_{nit} > U_{njt}$ $\forall i \neq j$ for any $i$ and $j$. Standard specification of $V_{njt}$ is constructed to be linear with product attributes [31]. For this study, we specify the utility function of a consumer $n$, selecting the sausage product $j$ ($j = 1, 2, 3$, for product 1, product 2 and neither) in the choice set $t$ ($t = 1, 2, 3, \dots, 12$), as below:

$$U_{njt} = \beta_0 + \beta_1 Price_{njt} + \beta_2 Meat_{njt} + \beta_3 Om3_{njt} + \beta_4 Rfat_{njt} + \beta_5 Rsalt_{njt} + \beta_6 Om3_{njt} * Rfat_{njt} + \beta_7 Om3_{njt} * Rsalt_{njt} + \beta_8 Rfat_{njt} * Rsalt_{njt} + \varepsilon_{njt} \tag{2}$$

Dependent variables were dichotomous, where 1 meant the alternative being chosen and 0 otherwise. The constant $\beta_0$ captured the effect of the opt-out option and represented the utility level if a consumer chose "neither". Dummy coding was adopted for a lower likelihood of misinterpretation [32]. Both main effects and specific two-way interaction effects were evaluated in the model. Nutrition claims regarding omega-3 addition, reduced fat and

reduced salt (corresponding to $Om3_{jt}$, $Rfat_{jt}$ and $Rsalt_{jt}$ in the model, respectively) were all binary variables, where 1 indicated the sausage product having a claim and 0 otherwise respectively. $Om3_{njt}Rfat_{njt}$, $Om3_{njt}Rsalt_{njt}$, $Rfat_{njt}Rsalt_{njt}$ were interactive variables.

Previous studies suggest that consumers have heterogeneous preferences to meat products and nutritional claims are of unequal value to different consumers [33–35]. Therefore, coefficients of observed variables should be allowed to vary among participants. A highly flexible logit model, namely a random parameter logit (RPL) model, was used for data analysis. The coefficient of price in the utility function is specified to follow a lognormal distribution, which is suitable where higher prices are consistently valued negatively [30,36]. The coefficients of non-price attributes, including the meat content and nutrition claims, could logically be of either sign, and are estimated independently as random parameters with a normal distribution. Each consumer is treated to a set of specific parameters reflecting individual preferences.

A generalized mixed model was estimated in WTP space to obtain consumers' WTP. The superiority of WTP space lies within plausible WTP estimates, and a better goodness of fit in the data [37,38]. The Equation (2) is re-parameterized in such a way that the coefficients directly represent marginal WTP for attributes and the prior assumptions of distributions are made with regard to WTP [37]. WTPs for nonprice attributes are specified to be normally distributed, as consumers' WTP could logically be either positive or negative. Equation (2) is re-parameterized as follows.

$$U_{njt} = \beta_{n0}/\mu_n - \lambda_n Price_{njt} + \lambda_n WTP_n X_{njt} \tag{3}$$

where $\mu_n$ is a scale parameter. $\beta_{nm}$ denotes the coefficient vector of the specified non-price attribute $X_{njt}$ (i.e., $Meat_{njt}$, $Om3_{njt}$, $Rfat_{njt}$, $Rsalt_{njt}$). The utility coefficients are defined as $\lambda_n = \beta_{n1}/\mu_n$, $c_n = \beta_{nm}/\mu_n$ and $WTP_n = -c_n/\lambda_n$. Only main effects were evaluated in the model.

### 2.4. Statistical Analysis

Paired sample T-tests were performed to analyze sensory liking data with different samples (conventional vs. omega-3 samples in the blind condition) and under information conditions (before vs. after the information disclosure of omega-3 fatty acid). Estimation of RPL and WTP parameters was achieved through the simulated log-likelihood method [30,36,37], using 2000 Halton draws. All data analyses were run by Stata 17 software (StataCorp. 2021. *Stata Statistical Software: Release 17.* StataCorp LLC., College Station, TX, USA).

## 3. Results

### 3.1. Participants

Table 2 reports participants' socio-economic and demographic characteristics. According to Table 1, 57% of the participants were female, 62% were between 18 and 24 years old, and 67.18% had a Bachelor's degree or above. Overall, the sample population was biased towards young and highly educated participants, with students accounting for a large proportion (74.23%). These characteristics are common in volunteer-based food-related consumer studies conducted on a university campus, especially when sensory evaluations are also involved [13,39].

The statistics for self-assessed familiarity with omega-3 ratings showed that more than half of the participants claimed to be very familiar with omega-3 (55%) and only 29% of participants expressed considerable unfamiliarity. Regarding consumption habits, 79% of participants ate processed meat at least once a week.

**Table 2.** Participants' socio-economic and demographic characteristics. Note: [1] About twenty-five per cent of respondents did not know or preferred not to answer the level of household income, who were assigned an average income level for further analysis. [2] Omega-3 familiarity was self-assessed using a structured 9-point hedonic scale, where 1 = "extremely unfamiliar", 5 = "neither unfamiliar nor familiar", and 9 = "extremely familiar".

| Variable | Category | Frequency | Percentage (%) |
|---|---|---|---|
| Gender | Male | 141 | 43 |
| | Female | 185 | 57 |
| | 18–24 | 202 | 62 |
| Age class | 25–34 | 86 | 26 |
| | 35–44 | 23 | 7 |
| | 45 and Over | 15 | 5 |
| Education Level | Secondary or less | 38 | 12 |
| | College credit, no degree | 65 | 20 |
| | Bachelor | 92 | 28 |
| | Master or professional | 102 | 31 |
| | Doctoral or above | 25 | 8 |
| | Others | 4 | 1 |
| | Student | 242 | 74 |
| Employment Status | Employed Full-Time | 64 | 20 |
| | Employed Part-Time | 16 | 5 |
| | Not Employed | 4 | 1 |
| | € 15,000 and below | 26 | 8 |
| | € 15,001–€ 40,000 | 91 | 28 |
| Household Income range | € 40,001–€ 80,000 | 62 | 19 |
| | € 80,001 and above | 64 | 20 |
| | Don't know or prefer no answer[1] | 83 | 25 |
| Familiarity with omega-3 ratings [2] | 1–4 | 94 | 29 |
| | 5 | 52 | 16 |
| | 6–9 | 180 | 55 |
| | 15 or more times a week | 12 | 4 |
| Eating frequency of processed meat (e.g., sausages, nuggets, burger, ham, bacon) | 7–14 times a week | 47 | 14 |
| | 4–6 times a week | 74 | 23 |
| | 1–3 times a week | 126 | 39 |
| | Less than once in a week | 67 | 21 |

*3.2. Sensory Liking Results*

Sensory liking ratings for sausages are illustrated in Figure 1. Under the blind condition, the conventional sausages scored significantly higher than omega-3 sausages regarding taste, texture, and overall liking. Similarly, the informed omega-3 sausages scored significantly higher than that of blind omega-3 sausages, regarding appearance, texture, and overall liking. Consumers held slightly positive attitudes toward omega-3-enriched sausages, where mean ratings of appearance, taste, texture, and overall liking were all between 5 (5 = "neither like nor dislike") and 6 (6 = "like slightly"). This finding agrees with some studies that omega-3 enriched sausages could be produced with appealing sensory properties [3,12,40].

*3.3. RPL Regression Results and WTP Estimation*

Table 3 lists the results of RPL model and WTP estimation. Results indicated that participants' utility and payment intentions were conjointly affected by selected attributes. Standard deviations for almost all estimated parameters were of statistical significance, meaning preference heterogeneity for selected attributes within the sample population, so that the RPL model used was more suitable in comparison to a standard logit model [30,36].

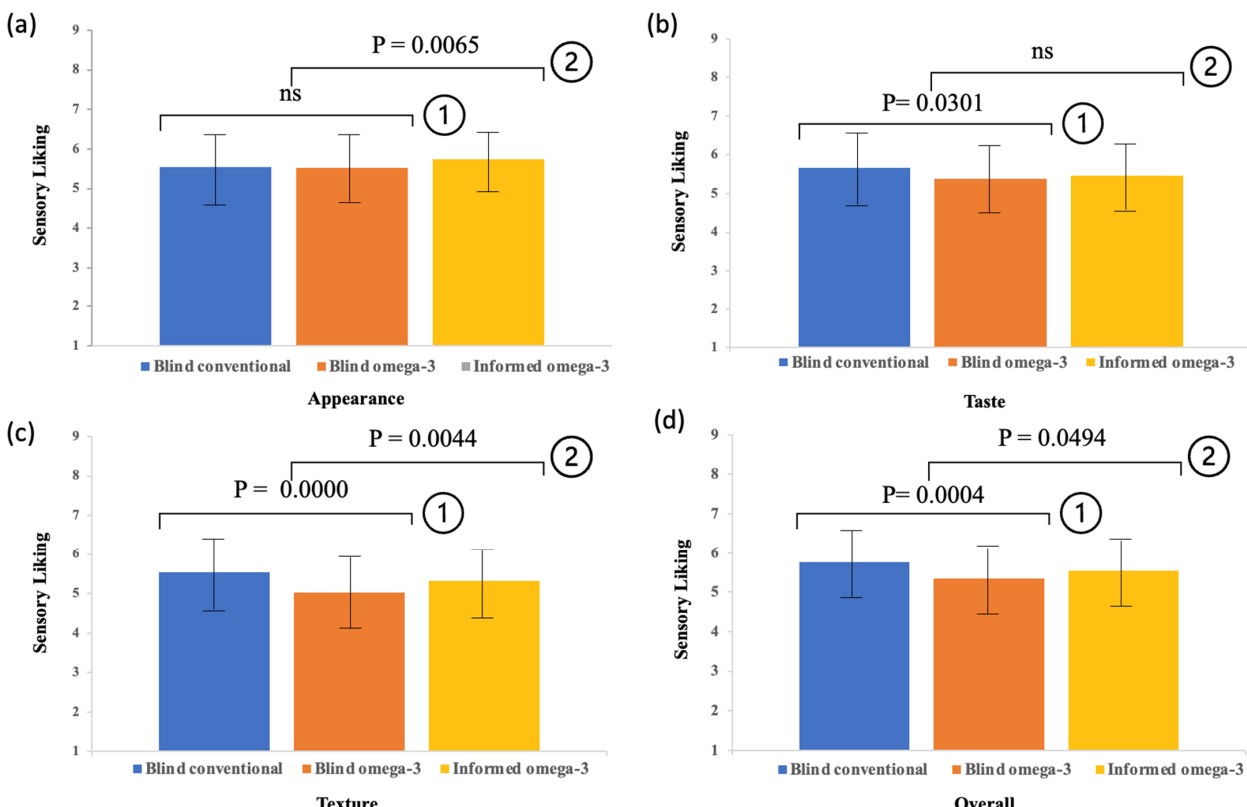

**Figure 1.** Mean ratings of (**a**) appearance, (**b**) taste, (**c**) texture and (**d**) overall liking are shown. Error bars represent standard deviation of the mean. ① = liking of conventional vs. omega-3 samples in the blind condition (paired Student's *t*-test); ② = liking before vs. after the information disclosure of omega-3 samples (paired Student's *t*-test).

Based on RPL results on the left-hand side of Table 3, a negative *Constant (neither)* coefficient indicated that consumer preferred other alternatives in comparison with the opt-out option in a choice set. As expected, a negative *Price* coefficient suggested an inverse relationship between payment and utility. Moreover, significantly positive coefficients of three specified attributes, namely *Meat*, *Om3 claim* and *Rfat claim*, indicated that consumers preferred pork sausages with a higher meat content, an omega-3 nutrition claim and a reduced fat nutrition claim. When only considering main effects, the coefficient of *Rsalt claim* was not significant, meaning that pork sausages with less salt were not preferred from those with regular salt. However, when considering the two-way interaction effects, the two interactive coefficients of *Om3*Rsalt* and *Rfat*Rsalt* were both significantly positive, meaning that consumers' preference for reduced salt pork sausages significantly increased when reduced salt pork sausages simultaneously had omega-3 addition or fat reduction. Therefore, more utility was obtained by consumers from pork sausages with the reformulation of having a higher meat content, an omega-3 nutrition claim, a reduced fat nutrition claim, a coexistence of an omega-3 addition and salt reduction claim, and a coexistence of a fat and salt reduction claim. Moreover, consumers welcomed healthier pork sausages reformulated by both nutrient "addition" and ingredient "reduction" strategies.

The right-hand side of Table 3 reports WTP estimation. Overall, consumers were willing to pay more for both "addition" and "reduction" reformulation, where carrying an omega-3 nutrition claim elicited the highest monetary reward from consumers and a reduced salt claim elicited the lowest. More specifically, on average, a consumer was willing to pay 0.52 euro for an omega-3 claim made on a pack of 454 g pork sausages, followed by 0.50 euro for a reduced fat claim, and 0.41 euro for a reduced salt claim. Furthermore, consumers were also willing to pay marginally more for higher meat content in sausages. On average, consumers' WTP for a 10% increase of meat content was estimated

to be 0.283 euro. These results were in line with findings in other studies that consumers demand healthier processed meat products [23,41–43] and they are willing to pay more for salt-reduced and fat-reduced processed meat products [6,44,45].

**Table 3.** Results of the RPL model and WTP estimation.

| Variable | RPL Model | | WTP Estimation | |
|---|---|---|---|---|
| | Coefficient | SD | Mean | SD |
| Constant (neither) | −9.13 *** | 6.70 *** | −2.61 *** | 2.03 *** |
| Price [1] | −2.30 *** | 1.28 *** | −3.21 *** | 2.55 *** |
| Meat [2] | 7.49 *** | 7.42 *** | 2.83 *** | 2.88 *** |
| Om3 claim | 0.91 *** | 0.77 *** | 0.52 *** | 0.51 *** |
| Rfat claim | 0.62 ** | 0.81 *** | 0.50 *** | 0.46 *** |
| Rsalt claim | −0.13 | 0.11 | 0.41 *** | 0.36 *** |
| Om3*Rfat | −0.21 | 1.00 *** | | |
| Om3*Rsalt | 0.92 ** | 1.77 *** | | |
| Rfat*Rsalt | 1.31 *** | 1.14 *** | | |
| Log-likelihood | −2585.83 | | −2626.92 | |
| Wald Chi-Square | 821.83 | | 1657 | |
| AIC [3] | 5207.65 | | 5277.84 | |
| No. of respondents | 326 | | 326 | |
| No. of observations | 11,736 | | 11,736 | |

Note: **, and *** denoted significance at the 5%, and 1% levels, respectively. [1] As described above, the price coefficients were assumed to be log-normally distributed, so the original mean and standard deviation for the log of the price coefficients were adjusted accordingly and reported as normally distributed coefficients. [2] When interpreting the coefficient of "meat content", the coefficient should multiply by 0.1, indicating a 10% increase/decrease of meat content. [3] AIC was short for Akaike information criterion.

### 3.4. WTP Estimation by Satisfied and Unsatisfied Sensory Preference Groups

A novel element of this study was the collection of sensory liking data on the physical prototype of omega-3-enriched sausages prior to collecting data on consumers' WTP in a CBC experiment. Therefore, based on the overall liking of omega-3 sausages, the total participants were classified into two groups: an unsatisfied sensory group (147 consumers) who gave overall liking ratings of one to five and a satisfied sensory group (179 consumers) who gave overall liking ratings of six to nine. WTP estimation by the two sensory preference groups is listed in Table 4.

The unsatisfied sensory group indicated more WTP for higher meat content (0.376 euro) than that from the satisfied sensory group (0.303 euro), which suggested that when consumers were not satisfied with their sensory experiences with omega-3 sausages, they placed more value on the meat content attribute. In contrast, WTP for an omega-3 nutrition claim from the unsatisfied sensory group (0.35 euro) was significantly lower than that from the satisfied sensory group (0.66 euro), which implied a relation between a higher level of sensory preference and more WTP for omega-3 enriched sausages. Apart from meat content and omega-3 claim, consumers' WTPs for pork sausages having a reduced fat claim and having a reduced salt claim were very similar between two different sensory preference groups, which indicated that unpleasant sensory experiences with omega-3 sausages had little collateral influence on consumers' valuation on sausages reformulated by "reduction" strategies. Standard deviations were larger for the unsatisfied sensory group than the other group, meaning that more variation was observed in consumers who had unsatisfied sensory experiences.

### 3.5. WTP Estimation by Omega-3 Unfamiliar and Familiar Groups

The participants' self-assessed familiarity ratings for omega-3 fatty acids were collected before the sensory test with the informed omega-3 sausages, and results (Table 2) indicated that more participants claimed to be familiar with the omega-3 than to be unfamiliar. To investigate influence of omega-3 familiarity on consumers' WTP, the total participants were classified into two groups: an unfamiliar group (146 consumers) who gave overall liking ratings of one to five and a familiar group (180 consumers) who gave overall liking ratings of six to nine. WTP estimation by the groups of two familiarity levels is listed in Table 5.

**Table 4.** WTP estimation by the satisfactory and unsatisfactory groups.

| Variable | Unsatisfied Sensory Group (N = 147) [1] | | Satisfied Sensory Group (N = 179) [2] | | *p*-Value [3] |
|---|---|---|---|---|---|
| | **Mean** | **SD** | **Mean** | **SD** | |
| Constant (neither) | −2.78 *** | 2.51 *** | −2.66 *** | 1.68 *** | |
| Meat [4] | 3.76 *** | 3.01 *** | 3.03 *** | 2.71 *** | 0.023 ** |
| Om3 claim | 0.35 *** | 0.65 *** | 0.66 *** | 0.36 *** | 0.000 *** |
| Rfat claim | 0.54 *** | 0.61 *** | 0.53 *** | 0.41 *** | 0.865 |
| Rsalt claim | 0.42 *** | 0.48 *** | 0.41 *** | 0.31 *** | 0.827 |
| Log-likelihood | −1226.98 | | −1376.60 | | |
| Wald Chi-Square | 596.42 | | 733.61 | | |
| AIC [5] | 2477.96 | | 2777.20 | | |
| No. of respondents | 147 | | 179 | | |
| No. of observations | 5292 | | 6444 | | |

Note: **, and *** denoted significance at the 5%, and 1% levels, respectively. SD is short for standard deviation. Sensory liking on sausages was evaluated using a horizontal Likert 9-point scale, where 1 = "dislike extremely", 2 = "dislike very much", 3 = "dislike moderately", 4 = "dislike slightly", 5 = "neither like nor dislike", 6 = "like slightly", 7 = "like moderately", 8 = "like very much", and 9 = "like extremely". [1] A unsatisfied sensory group indicated overall sensory liking ratings of 1 to 5. ". [2] A satisfied sensory group indicated overall sensory liking ratings of 6 to 9. [3] *p*-values were calculated from a two-sample z-test with Ho suggesting a WTP mean difference between the unsatisfied and satisfied sensory groups equaling zero. [4] When interpreting coefficient of "meat content", the coefficient should multiply 0.1, indicating a 10% increase/decrease of meat content. [5] AIC was short for Akaike information criterion.

**Table 5.** WTP estimation by the omega-3-unfamiliar and -familiar groups.

| Variable | Omega-3 Unfamiliar Group (N = 146) [1] | | Omega-3 Familiar Group (N = 180) [2] | | *p*-Value [3] |
|---|---|---|---|---|---|
| | **Mean** | **SD** | **Mean** | **SD** | |
| Constant (neither) | −3.17 *** | 2.24 *** | −3.02 *** | 3.43 *** | |
| Meat [4] | 2.70 *** | 2.63 *** | 3.45 *** | 3.21 *** | 0.010 ** |
| Om3 claim | 0.46 *** | 0.56 *** | 0.58 *** | 0.48 *** | 0.020 ** |
| Rfat claim | 0.51 *** | 0.47 *** | 0.49 *** | 0.54 *** | 0.721 |
| Rsalt claim | 0.48 *** | 0.38 *** | 0.40 *** | 0.38 *** | 0.059 |
| Log-likelihood | −1192.6727 | | −1426.8732 | | |
| Wald Chi-Square | 593.13 | | 502.59 | | |
| AIC [5] | 2409.345 | | 2877.746 | | |
| No. of respondents | 146 | | 180 | | |
| No. of observations | 5256 | | 6480 | | |

Note: **, and *** denote significance at the 5%, and 1% levels, respectively. SD is short for standard deviation. Self-assessed familiarity with omega-3 was evaluated using a horizontal Likert 9-point scale, where 1 = "extremely unfamiliar", 5 = "neither unfamiliar nor familiar", and 9 = "extremely familiar". Correlation between omega-3 familiarity rating and overall sensory liking ratings was low (0.0599). [1] A unfamiliar group indicated ratings of 1 to 5. [2] A familiar group indicated ratings of 6 to 9. [3] *p*-values were calculated from a two-sample z-test with Ho suggesting a WTP mean difference between the unfamiliar and familiar groups equaling zero. [4] When interpreting coefficient of "meat content", the coefficient should multiply 0.1, indicating a 10% increase/decrease of meat content. [5] AIC was short for Akaike information criterion. Note: [1] About twenty-five per cent of respondents did not know or preferred not to answer the level of household income, which were assigned an average income level for further analysis. [2] Omega-3 familiarity was self-assessed using a structured 9-point hedonic scale, where 1 = "extremely unfamiliar", 5 = "neither unfamiliar nor familiar", and 9 = "extremely familiar".

The omega-3-unfamiliar group indicated less WTP for higher meat content (0.270 euro) and an omega-3 nutrition claim (0.46 euro) than that from the familiar group (0.345 euro and 0.58 euro, respectively). This result shows that consumers who were more familiar with omega-3 were on average willing to give higher payment premiums than their counterparts who were unfamiliar. Similarly, with the satisfactory and unsatisfactory groups, consumers' WTP for a reduced fat claim and a reduced salt claim were not statistically different between the unfamiliar and familiar groups. Hence, the influence of omega-3 familiarity affected consumers' evaluation of omega-3 sausages, not on sausages reformulated by "reduction" strategies.

## 4. Discussion

This study used a sensory evaluation before measuring WTP for healthier pork sausages, with an aim of investigating existing ambiguities on consumer preferences for two opposing reformulation strategies (i.e., nutrient "addition" and ingredient "reduction") in healthier processed meat products. Results suggested that consumers should be willing to pay higher prices for both "addition" and "reduction" strategies (Table 3) in sausages reformulated to be healthier. The average highest WTP was for an omega-3 claim (under the "addition" strategy category), followed by a reduced fat claim and a reduced salt claim (both under the "reduction" strategy category), and the lowest was for a higher meat content. This challenges a large body of studies, finding that consumers place more value on "reduction" and little value on "addition" [6,8,24,42,46]. Therefore, this study highlights the importance of incorporating a sensory evaluation in the research process. Studies only using survey techniques are useful under the context that consumers often make food-purchasing decisions in the absence of having tasted a given food product. However, consumer perceptions and WTP for innovative processed meat products could be altered by the real sensory experience. Grunert, Verbeke, Kugler, Saeed and Scholderer [18] point out that judgment after experiencing novel meat products is more credible. Prior literature has provided inconsistent conclusions on consumers' perceptions of reformulated processed meat from studies with and without a tasting experience. For instance, using a survey, Schnettler, Ares, Sepulveda, Bravo, Villalobos, Hueche and Lobos [6] found that, on average, consumers are not willing to pay more for reformulated sausages carrying a fiber claim. In contrast, when incorporating sensory evaluations, Huber, et al. [47], Grasso, Monahan, Hutchings and Brunton [13], Diaz-Vela, Totosaus, Escalona-Buendia and Perez-Chabela [2] and Grasso, Rondoni, Bari, Smith and Mansilla [15] concluded that there are market prospects for fiber-enriched chicken burgers, sterol-enriched turkey, fiber-added sausages and vegetable-blended beef burgers, respectively. Moreover, the WTP estimates for nutrition claims of "reduced fat" and "reduced salt" found in this study are similar to Romagny, Ginon and Salles [44]'s findings of an approximately 12% payment increase for sausages reduced in fat and salt in a home-tasting environment. This is in contrast to Schnettler, Ares, Sepulveda, Bravo, Villalobos, Hueche and Lobos [6]'s research on sausages in a hypothetical setting, which concludes that claims of sodium and fat reduction correspond to marginal price increments of 1.2% and 5.6%, respectively.

This study reverses the normal procedure of a concept test followed by a prototype test in terms of developing novel meat products [18], but is appropriate in investigating how sensory liking affects the WTP for reformulated sausages and can more realistically explore consumers' perceptions of healthier processed meat. A sensory evaluation helps to reduce hypothetical bias toward healthier meat products and a satisfactory tasting experience helps to overcome barriers to consumption [9]. Results (Table 4) shows that the WTP for omega-3 sausages was significantly different between groups with satisfied and unsatisfied sensory experiences, where consumers who indicated higher liking ratings were on average willing to give higher payment premiums than their counterparts who indicated lower liking ratings. In line with many prior studies, sensory preferences are repeatedly confirmed as one of the most influential factors in purchasing and paying intentions for healthier meat products [23,42,48–51] and are positively correlated with consumers' WTP magnitudes for healthier sausages [5,44]. Therefore, compromise on sensory characteristics in exchange for more healthiness in processed meat products may lead to a lower price premium paid by consumers, as Romagny, Ginon and Salles [44] suggest that when consumers give lower pleasantness scores in comparison with the non-reformulated version, they are only willing to pay the same price for the reformulated sausages as for the original sausages. Notably, WTP for salt and fat reduction from the unsatisfied sensory group was similarly comparable with that from the satisfied sensory group. Hence, unpleasant sensory experiences regarding sausages reformulated by "addition" strategies had little collateral influence on consumers' valuation on sausages reformulated by "reduction" strategies. Furthermore, multiple healthier reformulations of nutrient "addition" and ingredient "reduction" made

on sausages could elicit higher utilities than only one reformulation, which agrees with Barone, Banovic, Asioli, Wallace, Ruiz-Capillas and Grasso [51]'s finding that together with fat and salt reduction, healthier meat products of plant-based ingredients are more acceptable to consumers. The influence of omega-3 familiarity levels on consumers' WTP for the omega-3-enriched, fat-reduced, and salt-reduced sausages was also investigated. The results (Table 5) implied a relation between a higher level of omega-3 familiarity and more WTP for omega-3-enriched sausages but had no effect on WTP for fat-reduced or salt-reduced sausages. Hence, there is evidence that consumer perceptions and payment intentions are influenced by their familiarity with the reformulated ingredient of the "addition" strategy. In line with Lahteenmaki's studies, unfamiliarity with the nutrient's health-related claim can negatively impact consumers' perceived healthiness of novel foods [52], but when familiar, a claim on the nutrient could increase perceived healthiness and consumer acceptance [53].

This study also found supporting evidence that consumers' sensory perceptions could be altered by the disclosure of health-related information. In Figure 1, the informed omega-3 sausages were significantly preferred over the blind omega-3 sausages on all sensory attributes except for taste. However, the average taste liking rating for the informed omega-3 sausages (5.47 scores) was numerically higher than that of blind omega-3 sausages (5.38 scores), but the difference was not statistically significantly. In line with findings of some other similar studies, giving consumers health information does not significantly change consumers' palate and the liking of healthier processed meat products, but it can help to mitigate some unliked sensory perceptions [11,13]. Furthermore, some studies also suggest a health-related claim increases consumers' WTP compared to when these products were presented without such information [13,15,54].

## 5. Conclusions

By carrying out a prototype sensory evaluation followed by a conjoint experiment, this study found consumers were willing to pay more for both reformulation strategies of "addition" and "reduction" in pork sausages. Sensory preferences were an influential factor in WTP, where WTP was positively correlated with sensory liking levels. These findings provide important implications for food manufacturers. Although pork sausages are often considered to have an unhealthy food image, consumers still welcome healthier sausages and are willing to pay premiums for both nutrient "addition" and ingredient "reduction" reformulation. Notably, an effective marketing approach is to offer opportunities for consumers to taste the newly reformulated processed meat products, to inform consumers of health-related reformulation information and to continuously optimize the consumers' sensory experience, especially when many consumers may find it difficult to relate to new launches of nutrient added processed meat products.

*Limitation and Future Research*

The limitations to the present study are two-fold. One is that the consumer sample was not a true representation of the general population. Therefore, the conclusions are more applicable to young consumers with high levels of education. The other is the potential overestimation of WTP due to using a stated preference approach. Future research is recommended to test a wider range of healthier meat prototypes with representative survey samples. In this study, we argue that incorporating a sensory evaluation produced a more realistic measurement of consumers' perceptions of two meat reformulation strategies. To further confirm the effect of a sensory evaluation in reducing hypothetical bias and skepticism toward healthier meat products, a comparison study should be conducted to verify this inference by allowing one group do a sensory evaluation and not the other group, and then comparing the consumption intentions between the two groups. Non-hypothetical experiments, such as auctions with real money transactions, could potentially calibrate the implicit over-stated payment intentions.

**Author Contributions:** Conceptualization, X.H., C.L. and F.J.M.; methodology, X.H., C.L. and F.J.M.; software, X.H. and C.L.; validation, C.L.; formal analysis, X.H.; investigation, X.H. and F.J.M.; resources, C.L., L.W. and F.J.M.; data curation, X.H., C.L. and F.J.M.; writing—original draft preparation, X.H., C.L., L.W. and F.J.M.; writing—review and editing, X.H., F.J.M., S.G. and M.W.; visualization, X.H., S.G. and M.W.; supervision, C.L., L.W. and F.J.M.; project administration, X.H., C.L., L.W. and F.J.M.; funding acquisition, L.W. and F.J.M. All authors have read and agreed to the published version of the manuscript.

**Funding:** This research was funded by the Food Institutional Research Measure of the Irish Department of Agriculture, Food and the Marine grant number Project 11/F/035.

**Institutional Review Board Statement:** The study was conducted in accordance with the Declaration of Helsinki, and approved by the Ethics Committee of University College Dublin (protocol code LS-17-91-Hong-Li and 15 January 2018).

**Informed Consent Statement:** Informed consent was obtained from all subjects involved in the study.

**Data Availability Statement:** The data presented in this study are available on request from the corresponding author.

**Acknowledgments:** The authors would like to thank Zhifeng Gao and Emma Feeney for their valuable opinions on the research design, Vincenzo del Grippo his unparalleled help with preparing lab-made sausages, and Si Wu, Rao Fu, Yujie Shi for their great assistance in executing the sensory evaluation.

**Conflicts of Interest:** The authors declare no conflict of interest.

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
