# Peer review of "Consumer Preferences for Processed Meat Reformulation Strategies: A Prototype for Sensory Evaluation Combined with a Choice-Based Conjoint Experiment"

_agriculture, doi:10.3390/agriculture13020234_

Round 1

Reviewer 1 Report

The paper “Consumer preferences for meat reformulation strategies: A prototype for sensory evaluation combined with a choice-based conjoint experiment evaluates consumers' willingness to pay for adding omega-3” is fascinating. However, the research topic is sensory evaluation and willingness to pay. The design of sensory assessment is not rigorous. I hope the author can answer and supplement my question before even being considered for publication.

Methods

Lines 138-148: The explanation of Table 1 should be placed in the Result section rather than the Methods section.

Lines 174-178: Is it appropriate to place the purpose of sensory evaluation in the Methods section? It is suggested to put the purpose in the Introduction section and rewrite it.

Line 248 Research papers should use objective expressions.

Formula (1), (2), and (3) have duplicate labels, please delete and correct them.

Why is the title of 2.3 not sensory evaluation? I noticed that this section focuses on sensory evaluation rather than data collection. It is necessary to separate sensory evaluation from this section.

At the same time, many problems with sensory evaluation need to be clarified. How many samples are needed for the sensory evaluation experiment? How to control the variables of sensory samples, such as how much to reduce the salt and fat content, whether there are control samples, and whether omega-3 is added to the sausages with salt and fat reduction. Whether consumers taste the samples repeatedly during sensory evaluation. Please describe the sensory experiment in detail.

Similarly, I have doubts about the author's questionnaire. How can 24 of the 128 choices be screened? What are the cognitive needs? What are the 12 selection tasks? Please clarify this part.

Results

Table 3 has too few sensory attributes. At the same time, I did not see the difference between each product, which made me think that the design of sensory evaluation in this study was meaningless. Moreover, sensory evaluation in CBC is not shown in section 3.1,

Lines 274-275 Why is it attractive and pleasant? The general preferences of the three products are not different.

In general, the design of the sensory evaluation experiment is unreasonable and has enormous flaws.

Reviewer 2 Report

This paper reports the results of a study investigating consumer responses to novel pork sausage prototypes, developed to include omega-3 oil. The authors used a sensory evaluation in tandem with a choice-based conjoint experiment. Previous work has suggested conflicting influences of reduction and addition techniques to improve consumer willingness to buy processed meat products formulated to be healthier than standard products. This study aims to contribute to this understanding, as well as assess whether liking affects willingness to buy and whether informing consumers of the health aspects matters. 

Overall, the research protocol seems sound and the study includes a high N. The research question itself is interesting, and relevant for the journal. I have some comments for the authors, however: 

- the "hypothetical bias" is an interest topic, but I wonder whether there is room in the introduction to discuss the fact that often, in reality, consumers are making product choices in exactly this scenario i.e. without having previously tasted something they are considering purchasing. Does this not mean that hypothetical studies are also highly relevant to understanding willingness to buy?

- The authors have data on participants' familiarity with omega-3, so it might be interesting to assess whether the willingness to buy sausages containing omega-3 is influenced by this familiarity. In other food applications, familiarity is highly relevant - I think this would add value to the results section

- Why were a series of paired t-tests used instead of repeated measures ANOVA, with posthoc pairwise assessments adjusted for multiple comparisons? This seems like a more appropriate analysis.

- I think Table 3 should be replaced with a plot showing the results, this would make a more visually interesting results section (and separate the sensory evaluation results from the WTP results more clearly for the reader)

- ln 279: "proved" - in general, I think researchers should avoid strong terms such as "prove" in favor of softer words such as "provides evidence for", "implies", or "indicates"

- ln 287: negativity -> negative

- ln 312: "they intend to pay more for..." do they? Don't these results suggest that people might be more willing to pay for, instead of "intending" to pay? Intention feels like too strong a claim to me. 

- ln 346: "this study highlights the importance of incorporating a sensory evaluation in the research process to more accurately estimate...". Are the authors sure this approach gives a more "accurate" result? It seems to me that there is a difference between estimating willingness to pay for a product that a person has tasted vs one they haven't, and that one answer isn't more "accurate" than another, they simply reflect different contexts. I agree that sensory evaluations are of course important in this application and in product development more generally, but I think the authors are forgetting that consumers often make purchasing decisions in the absence of having tasted a given product. Finally, whether tasting products before the willingness to pay component of the study was run has primed participants in some way should perhaps be considered. 

- I wonder whether willingness to pay for omega-3 was similar to that for meat content, it seems that the authors don't mention this (I understand why, in that they want to compare addition vs reduction health claims). Nonetheless, it seems relevant to present and discuss which of the factors included in their regression analyses has the strongest and weakest effects in terms of WTP, rather than only compare across the three health-related factors.

- ln 373: "a positive correlation was observed between higher liking ratings and higher payment premiums" - was it? It seems to me that no correlation analysis was conducted to show this, rather Table 5 presents the results of separate analyses for two groups. I strongly suggest reformulating this part of the discussion section (as it seems inaccurate, the groups were not directly compared in this way nor were liking scores included as a factor in the WTP analysis) or reanalyzing the data in the way that would support this.

- Last paragraph of the discussion: I think it would be interesting if the authors discussed why "taste" was not affected by information provision. Given that they discuss earlier in this section the importance of a pleasant sensory experience, and that pleasant taste is likely important for repurchase of products, this seems valuable.

Reviewer 3 Report

Title - appropriate

Abstract – well written and adequate

Introduction the introduction is well written, the only suggestion is to cite the following reference (Hong, Xinyi, Chenguang Li, Liming Wang, Zhifeng Gao, Mansi Wang, Haikuang Zhang, and Frank J. Monahan. "The Effects of Nutrition and Health Claim Information on Consumers’ Sensory Preferences and Willingness to Pay." Foods 11, no. 21 (2022): 3460.).

Material and methods must be improved. The results are presented in the material and methods. It is necessary to cite Figure 1, which has already been published in the paper (Hong, Xinyi, Chenguang Li, Liming Wang, Zhifeng Gao, Mansi Wang, Haikuang Zhang, and Frank J. Monahan. "The Effects of Nutrition and Health Claim Information on Consumers' Sensory Preferences and Willingness to Pay." Foods 11, no. 21 (2022): 3460.)

Results are well presented, including tables and graphs. Sections from Materials and Methods related to results should be inserted in the appropriate place.

Conclusions are satisfactorily presented and come from the objectives and results of the research. The only thing I would suggest is to separate study limitations and future research from the conclusions. I think it should be written as a separate section within the discussion.

Round 2

Reviewer 1 Report

The paper has been revised. I suggest to accept it. 

Reviewer 3 Report

Dear Editor,

I am satisfied with the author's comments and corrections. Only if I saw correctly the part of limitations and future research remained as part of the conclusions. As I wrote, limitations and future research should be detailed in your Discussion section.
Just move it to the end of the discussion.
The work can be accepted in such a form after this correction.
